# A neuromorphic processor with on-chip learning for beyond-CMOS device integration

Hugh Greatorex [1,2] ✉, Ole Richter [3,4], Michele Mastella [5], Madison Cotteret [1,2,6], Philipp Klein [1,2], Maxime Fabre [1,2,7], Arianna Rubino [8,9], Willian Soares Girão[1,2], Junren Chen [9], Martin Ziegler[10], Laura Bégon-Lours [8], Giacomo Indiveri [9] & Elisabetta Chicca [1,2] ✉

Recent advances in memory technologies, devices, and materials have shown great potential for integration into neuromorphic electronic systems. However, a significant gap remains between the development of these materials and the realization of large-scale, fully functional systems. One key challenge is determining which devices and materials are best suited for specific functions and how they can be paired with complementary metal-oxide-semiconductor circuitry. To address this, we present a mixed-signal neuromorphic architecture designed to explore the integration of on-chip learning circuits and novel two- and three-terminal devices. The chip serves as a platform to bridge the gap between silicon-based neuromorphic computation and the latest advancements in emerging devices. In this paper, we demonstrate the readiness of the architecture for device integration through comprehensive measurements and simulations. The processor provides a practical system for testing bio-inspired learning algorithms alongside emerging devices, establishing a tangible link between brain-inspired computation and cutting-edge device research.

The energy requirements of current deep learning algorithms have promoted research into alternative computing architectures and technologies. Some of these efforts are aimed at emulating the computational principles of biological intelligence to enhance efficiency and processing capabilities. In this regard, the development of neuromorphic computing architectures has seen substantial growth[1–7]. In particular, neuromorphic systems using hybrid complementary metal-oxide-semiconductor (CMOS)-memristive circuits offer a promising direction for low-power, highly compact solutions where computation is performed in-memory[8,9]. Memristive technologies encompass a wide range of novel electronic materials and devices that possess inherent memory and reprogrammability through state-dependent, and usually non-volatile, resistance modulation[10,11].

When integrated in CMOS Spiking Neural Network (SNN) chips, hybrid neuromorphic/memristive circuits can exploit the physics of the devices and their intrinsic dynamics to carry out low-power computations that extend the advantages of conventional In-Memory Computing (IMC) dense crossbar array architectures[12]. IMC-based

[1]Bio-Inspired Circuits and Systems (BICS) Lab, Zernike Institute for Advanced Materials, University of Groningen, Groningen, The Netherlands. [2]Groningen Cognitive Systems and Materials Center (CogniGron), University of Groningen, Groningen, The Netherlands. [3]Asynchronous Integrated Circuits, Embedded Systems Engineering, DTU Compute, Technical University of Denmark, Copenhagen, Denmark. [4]Asynchronous VLSI and Architecture Group, School of Engineering & Applied Science (SEAS), Yale University, New Haven, CT, USA. [5]Neuronova Ltd., Milan, Italy. [6]Micro- and Nanoelectronic Systems (MNES), Technische Universität Ilmenau, Ilmenau, Germany. [7]Forschungszentrum Jülich, Jülich, Germany. [8]D-ITET Integrated Systems Laboratory, ETH Zurich, Zurich, Switzerland. [9]Institute of Neuroinformatics, University of Zurich and ETH Zurich, Zurich, Switzerland. [10]Energy Materials and Devices, Department of Materials Science, Kiel University, Kiel, Germany. ✉e-mail: h.r.greatorex@rug.nl; e.chicca@rug.nl

artificial neural network accelerators, which typically use either Static Random Access Memory (SRAM) or memristive crossbars, aim to maximize peak throughput, area, and power efficiency by circumventing the von Neumann bottleneck[13–16]. In contrast, mixed-signal neuromorphic architectures seek to further reduce overall power consumption, especially in edge computing applications like bio-signal processing or environmental monitoring, which involve slowly varying signals[17–19]. Recent research has focused on brain-inspired neural mechanisms to implement efficient neural networks targeting such applications[20–22]. These types of architectures implement SNNs, where the spikes are digital events communicated via asynchronous digital logic. The analog circuits implementing neural and synaptic dynamics, together with the asynchronous digital circuits handling event-based routing and network programmability, enable ultra-low-power computation. Typically, the analog circuits used in these neuromorphic platforms rely on the subthreshold analog transistor regime[23] to emulate neuron-like dynamics for a further reduction in energy cost[6,24,25].

By exploiting the physics of the CMOS devices, this approach has led to the development of a diverse array of circuits that implement computational models of synaptic plasticity[26]. Synaptic plasticity is the ability of synapses to be potentiated or depressed in a volatile (short-term plasticity) or non-volatile manner (long-term plasticity)[27]. Although pure CMOS hardware implementations of local synaptic plasticity rules have been shown to express complex and versatile computational properties[2,28,29], they require substantial silicon real-estate to store the synaptic weights. A common strategy for addressing this issue has been to drive the weight to a stable value for storage. The use of bistable plastic synapses originates from some of the first developments of full-scale neuromorphic systems[29,30] mimicking biological synapses which inherently have limited bit precision[31,32]. Other works propose to update the weights directly within a digital memory[2,33], thus facilitating long-term storage, but they often require a continuous power supply to maintain the memory. Combining the mixed-signal neuromorphic engineering approach with the integration of memristive devices, would simultaneously enable the exploration of additional computational strategies, such as intrinsic stochasticity and state-dependence, as well as provide a compact and non-volatile storage option for maintaining weight values during power-cycles.

Recent efforts have thus initiated the exploration of integrating memristive devices with CMOS neuromorphic systems, aiming to leverage the synergy of both technologies[34–40]. A majority of these efforts have focused on complementing memristive crossbar arrays with neuromorphic peripheral circuitry to handle the generation of output spikes and the computation of learning signals[35–37]. Synaptic weights in these systems are realized by the resistance states of memristive devices in a crossbar array. Such crossbars are effective for performing matrix-vector multiplications (MVMs) when the weights are programmed once and read multiple times during network inference. However, to perform real-time on-chip learning, high-frequency and temporally unstructured reprogramming of these devices is required while retaining weight reading functionality. Therefore, few works explore the possibility of implementing IMC-based synaptic plasticity, with learning directly occurring at each synaptic device. In[38] the authors proposed a differential three-terminal device interface to achieve more flexible device access for online learning while[39] and[40] proposed the exploitation of memristive device dynamics to implement in-memory plasticity directly in the crossbar. Although these approaches have been explored, they have been limited to simulations of a few circuit elements with restricted learning flexibility.

In this work, we introduce TEXEL, a fabricated chip combining the operational efficiencies of memristive devices with the spike-based approach of neuromorphic systems. The chip exploits the analog subthreshold CMOS regime and event-based computation to implement ultra-low power spiking neurons and plastic synapses with tunable always-on trace-based local learning functionality. TEXEL incorporates a Back-End Of Line (BEOL) device-agnostic differential synaptic interface, enabling the integration of a wide range of two- and three-terminal memristive devices across ~10K plastic synapses atop the CMOS chip. This design makes TEXEL a versatile research platform for large-scale BEOL device integration in neuromorphic systems. The platform addresses the research challenge of evaluating different memristive technologies within the context of a functional SNN processor, complementing ongoing efforts to overcome challenges in large-scale memristor fabrication and integration. While memristive devices are yet to be integrated, the processor exploits the synergies of IMC and SNNs to present a concrete step towards the following key developments for such systems:

- Exploiting the capabilities of memristive materials and devices to facilitate the implementation and consolidation of on-chip synaptic plasticity.
- Providing a platform to explore the large-scale BEOL integration of memristive materials and devices with an SNN processor.

Here, we present the TEXEL chip's architectural design and learning mechanisms within the context of neuromorphic computing and beyond-CMOS device integration. Through comparisons with other existing full-scale memristor-CMOS spiking neuromorphic processors, we describe TEXEL's design features and discuss its role in developing brain-inspired computing.

## Results
Silicon measurements that validate the functionality of the TEXEL chip (Fig. 1) are outlined in the following sections. Experiments using the on-chip learning circuits demonstrate the emergent Spike-Timing-Dependent Plasticity (STDP)[41] and Spike-Rate-Dependent Plasticity (SRDP)[42]. The functionality of the memristive device read-write circuits is verified experimentally, and the operation of the interfacing circuits is demonstrated with post-layout simulations, which define the parameter range of memristive devices aiming for compatibility with TEXEL. We present simulations of the CMOS device interface circuits with a Valence Change Mechanism (VCM) compact model to verify synaptic state switching and robustness of reading operations. Power consumption measurements provide a detailed breakdown of the contribution of each circuit block, exemplifying the inherent power efficiency advantages of subthreshold analog circuitry and the event-driven paradigm. Additionally, we perform network-level experiments and implement an on-chip SNN that demonstrates TEXEL's capabilities in neuromorphic computing applications utilizing Vector-Symbolic Architectures (VSAs). The chip builds upon previous work[25,38], moving beyond test configurations towards an integrated neuromorphic architecture with improved scalability. The device interface circuit provides enhanced flexibility and compatibility with different memristive technologies, addressing integration challenges in hybrid CMOS-memristive systems.

### Neural circuits
We measured the activation of the silicon neuron circuits to assess their transfer function and operating regimes. A Direct Current (DC) input was applied and systematically increased across all neurons while their spike rate was recorded (Fig. 2a). The resultant Frequency vs. Current (FI) curve shows both the aggregate mean response for each core as well as the individual activation profiles of all neurons. The discernible core-specific disparity is attributable to mismatch in the biasing circuitry. The dispersion in the FI curve of each neuron stems from inherent variations in individual neuron circuits. While device mismatch variability can be reduced by including calibration procedures for each element[2], we chose to minimize it, through judicious analog circuit design techniques, and keep it, as it can be exploited for example in learning[43].

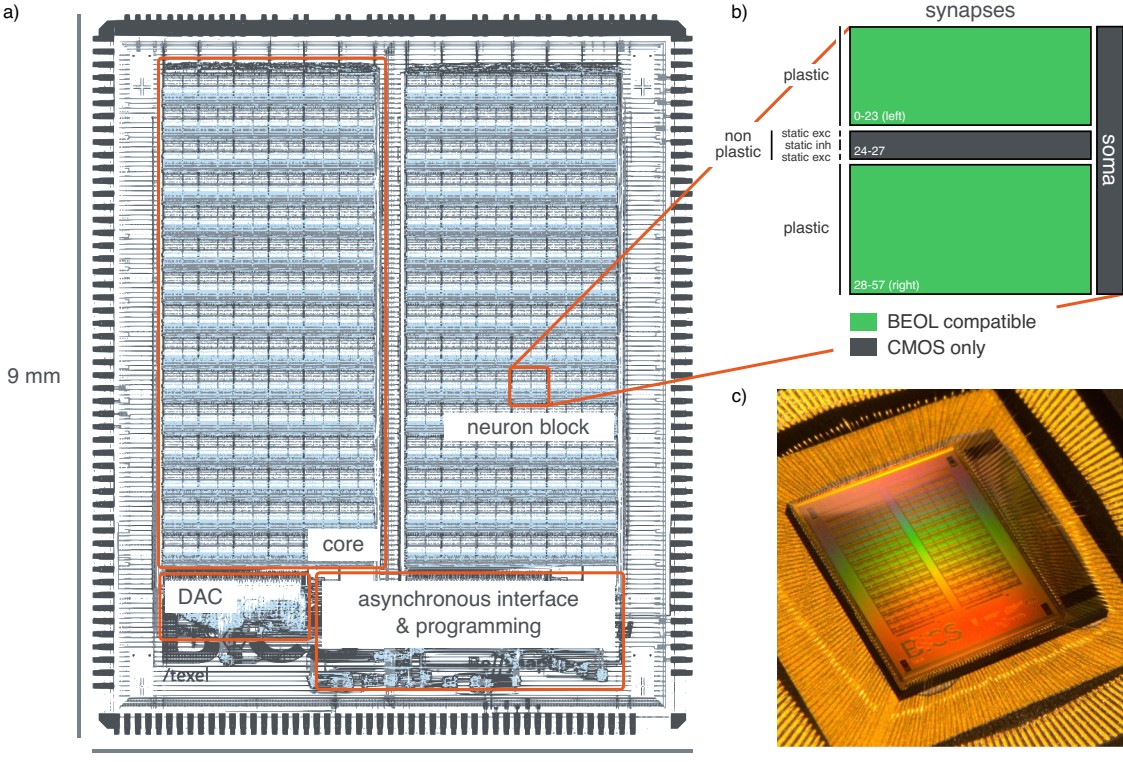

**Fig. 1 | The fabricated TEXEL chip. a** Footprint of the chip, indicating the location of the architectural blocks. **b** The neuron block footprint, indicating the synaptic fan-in of the soma within the block. The location of the plastic and non-plastic synapses is shown with excitatory (exc) and inhibitory (inh) types. The plastic synapses contain the contacts and interface circuitry for BEOL integration of memristive devices. **c** A photograph of the 9 mm × 7.5 mm die, fabricated with the X-FAB 180 nm process.

We conducted validation measurements of the adaptive characteristics embedded in the circuitry of each neuron. The response of the membrane potential to a DC step input was measured, as well as the timing of output spikes (Fig. 2b). These observations reveal the expected temporal pattern in the neuron's instantaneous spike rate, characterized by an initial peak followed by a gradual decay towards a stable state (Fig. 2c). Figure 2d shows an instance in which a neuron is stimulated by a Poisson spike train through its static excitatory synapses. The plastic synapses located within each neuron block are quantized to a binary value which is translated into an analog bias representing high and low synaptic efficacy. The on-chip weight matrix, encoding the state of all plastic synapses, can be read-out post learning and also programmed for inference (see Supplementary Fig. S5).

## Learning circuits

The on-chip plasticity was implemented using mixed-signal circuitry embedded within each plastic synapse. This circuitry emulates the Bistable Calcium-based Local Learning (BiCaLL) rule[44], which combines STDP for low activity with Hebbian changes[45] at high activity. The combination of both learning rules allows for different learning mechanisms to be explored in the context of an on-chip SNN, with STDP being advantageous for temporal pattern recognition and more efficient learning with sparse spiking, while SRDP excels at rate-based learning and evidence accumulation. In this model, synaptic updates are driven by pre- and postsynaptic calcium traces representing neuronal activity. A secondary postsynaptic trace ($Ca^{2+}$) with a slower time constant acts as a plasticity gating mechanism, ensuring weight updates occur only within specific firing rate ranges. The learning rule imposes a bistable analog internal weight ($V_w$) to help mitigate catastrophic forgetting in binary synapses[42,46–48], stabilizing synaptic states using accumulated updates and bistability circuitry.

Figure 3a shows measurements of a plastic synapse undergoing short-term depression. Pre-trace integration of presynaptic spikes maintains a decaying record of presynaptic activity, but without postsynaptic activity, the synaptic weight remains unchanged. When postsynaptic spikes occur, plasticity becomes apparent, and if the post-trace crosses its lower threshold, depression is triggered. The synaptic weight experiences short-term depression but stabilizes to the high state due to bistability circuitry.

We conducted in-silico experiments to characterize STDP of the learning circuitry (Fig. 3b, c). The synaptic weight changes ($\Delta w$) were measured by systematically varying pre- and postsynaptic spike timings. Adjusting the biasing parameters allowed for on-chip configuration of STDP curves, enabling the introduction of depressive regions for positive pre-post pairings. Additionally, SRDP was measured by varying pre- and postsynaptic Poisson spike rates. A probability map (Fig. 3d) of synaptic weight changes demonstrates that under conditions of high presynaptic and postsynaptic activity, the likelihood of the synapse settling into a potentiated (high) state increases. In contrast, when activity levels are lower, the synapse is more likely to undergo depression, favoring the low-weight state. This data highlights the sensitivity of the learning circuitry to the frequency and timing of local spiking activity.

## Memristive device interfacing circuits

Each plastic synapse on TEXEL (Fig. 1b) can be enabled to utilize a pair of memristive devices to store a binary weight using a differential device configuration[49,50]. When the chip is programmed to enable device operation, at the time of a presynaptic spike, the synaptic weight is read using a differential normalizer circuit[38]. To demonstrate the operation of the normalizer circuitry we performed extensive Spectre post-layout simulations over a range of

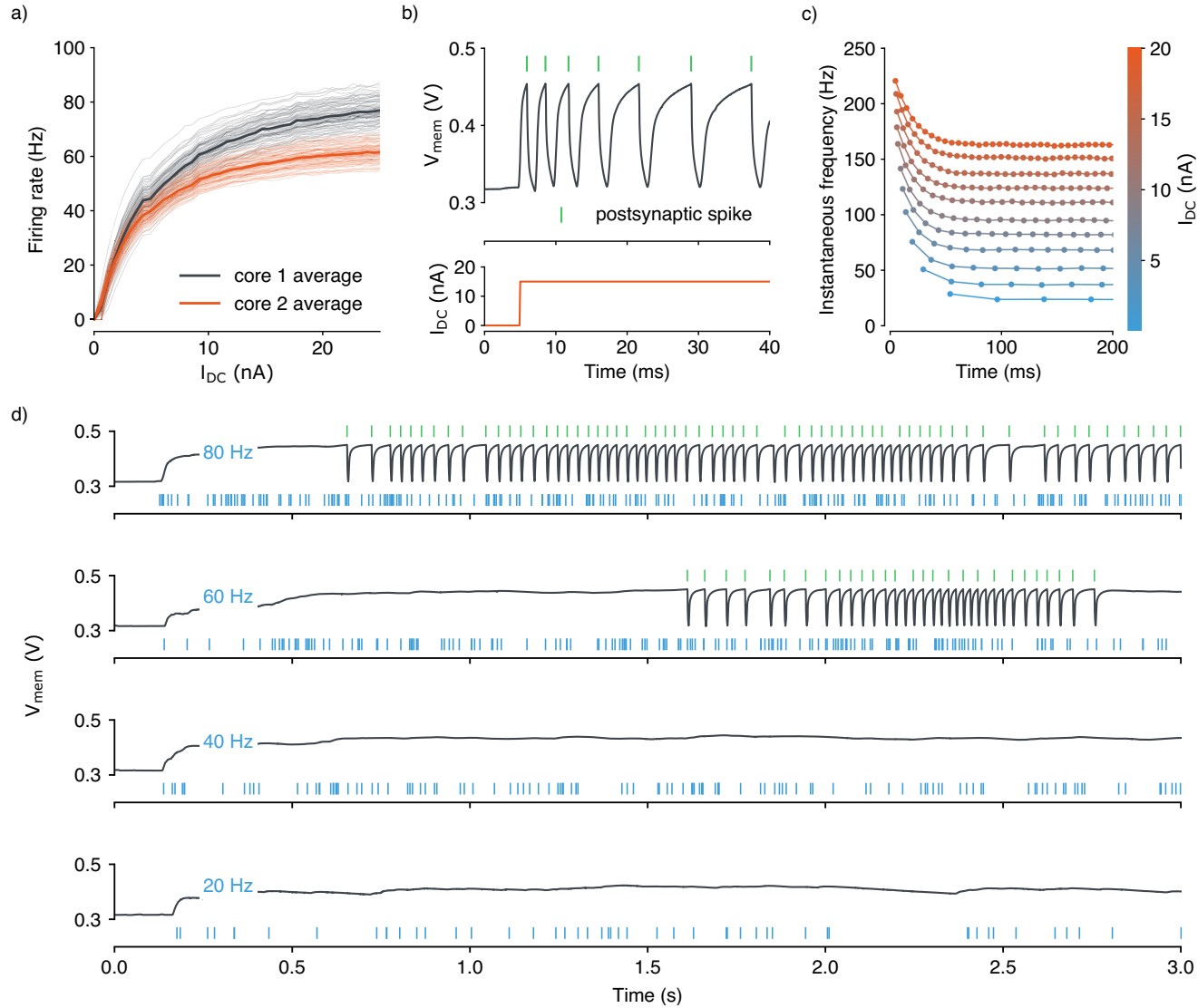

**Fig. 2 | Measurements of the neuron circuitry on the TEXEL chip. a** The measured firing rates of all neurons on TEXEL in response to a constant DC input current. **b** Measurement of the membrane potential of a single neuron in response to a DC step input. The neuron's adaptation characteristic is evident as its firing rate begins high and gradually diminishes to attain a steady state. **c** Measurements of the variations in the instantaneous firing rate and timing of output spikes in relation to the magnitude of DC injected into the soma. **d** The recorded membrane potential response of a neuron receiving presynaptic Poisson input at the static excitatory synapses. Below, in blue, is the presynaptic spike train, while above, in green, the postsynaptic spikes indicate the neuron's spiking activity.

memristive device parameters, namely: conductance, capacitance and on-off ratio. The memristive devices were modeled as parallel RC circuits. Figure 4a shows how the differential device setup, consisting of a "positive" and "negative" device, is able to store the binary synaptic weight. In the case where the resistance of the positive device is lower than that of the negative device, the current sourced through the positive device, $I_{pos}$, during a read pulse (presynaptic spike) is greater than the current sourced through the negative device, $I_{neg}$. In this scenario the normalizer circuit transmits a current, $I_{norm}$, proportional to the biasing of the normalizer circuit, norm_bias. For these simulations the current was normalized to 200 nA (norm_bias) and passed into a Differential Pair Integrator (DPI) synapse[51] to elicit a postsynaptic current, $I_{syn}$. In the alternative case, when the differential synapse is programmed to represent a low weight, the positive device resistance is greater than the negative device resistance. Therefore $I_{neg} > I_{pos}$ and the normalizer circuit does not convey a current. Figure 4b presents post-layout simulation results showing how $I_{norm}$ varies with the ratio of the positive and negative device conductance. When the ratio is <1,

$I_{norm}$ is zero, conversely when the ratio is >1, $I_{norm}$ is large enough to elicit a postsynaptic current. For large ratios between the positive and negative devices, translating to a large on-off ratio, the differential synapse and normalizer circuit is able to source a current that is closer to norm_bias.

The interfacing circuitry provides substantial protection against variations and drift in the integrated memristive devices. By employing a pair of memristive devices programmed in opposing states and connecting them to a normalizer circuit, we achieve a considerable reduction in output current variability. This configuration maintains robust performance as long as a sufficient difference between the High Resistance State (HRS) and Low Resistance State (LRS) is preserved. The impact of device drift can be assessed using Fig. 4c, where drift phenomena would manifest as horizontal movement on the graph, typically trending toward reduced $G_{on}/G_{off}$ ratios. This representation allows for empirical evaluation of how resistance state drift affects the differential normalizer synapse performance over time. The circuit's tolerance to such changes highlights its suitability for real-world applications where device stability remains challenging.

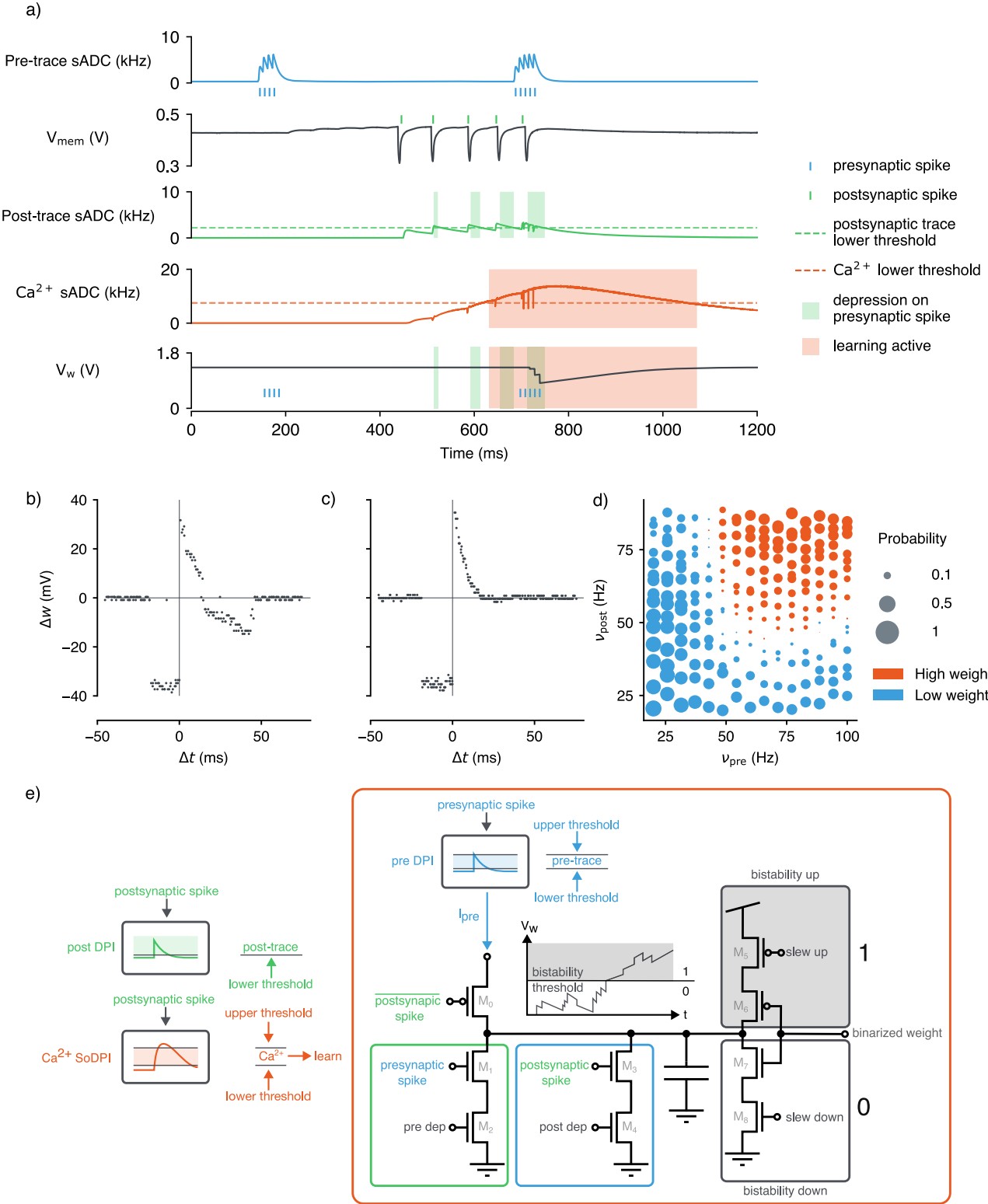

## Memristive device requirements

To quantify the compatibility of TEXEL with co-integrated memristive devices we performed extensive Spectre post-layout simulations of the CMOS interface circuitry with realistic device characteristics, over several orders of magnitude. We parameterized all simulations using a fixed read voltage pulse width of 500 μs and a norm_bias of 200 nA, however these can be varied using the on-chip programming and biasing. Figure 4c shows a heat-map of a 2D

logarithmic device characteristic sweep during which the on-off conductivity ratio of the device was varied with the on-resistance. This heat-map shows the percentage of the norm_bias of the normalizer circuit that was sourced during a read voltage pulse that was sent to the differential device synapse when storing a high weight. This is used as the metric determining whether a memristive device will operate as expected when integrated with the TEXEL chip and defines the compatibility. Similarly, we performed simulations

**Fig. 3 | Silicon measurements of a single plastic synapse and its neuron, demonstrating local synaptic plasticity. a** A presynaptic spike train induces a current (blue) read by the spiking Analog-to-Digital Converter (sADC), while simultaneous stimulation with an Excitatory Post Synaptic Current (EPSC) triggers postsynaptic spiking (green). Shaded regions indicate when the post-trace exceeds the lower threshold, reflecting short-term memory. The Ca²⁺ trace (orange) accumulates postsynaptic activity, showing plasticity when above its threshold. **b** STDP measurements assess the impact of pre- and postsynaptic spike timing, $\Delta t = \text{pre} - \text{post}$, on the analog weight of the synapse ($w$). This is the most expressive STDP curve achievable on-chip where all potentiation and depression branches of the learning circuitry are active. **c** This STDP curve demonstrates modulation of potentiation and depression through analog biasing. In this experiment the effect

of depression with positive $\Delta t$ timings was switched off. **d** SRDP results show the probability of the synapse maintaining high or low weight based on pre- and postsynaptic firing rates ($\nu_{pre}$ and $\nu_{post}$). **e** The plasticity circuit in each synapse uses three analog traces to govern weight updates. Two neuron-level traces, the post-synaptic trace (post-trace) and the Ca²⁺ trace, are transmitted to synapses and must meet threshold conditions for weight updates. If the post-trace exceeds a threshold, incoming presynaptic spikes reduce the synaptic weight by a fixed increment. The presynaptic activity (pre-trace) also determines whether the weight will increase or decrease, with updates occurring via charge deposition on a capacitor. The weight is then quantized into high or low states by a bistability circuit, which controls drift toward ground or supply voltage, with drift rates set by the slew up and slew down biases.

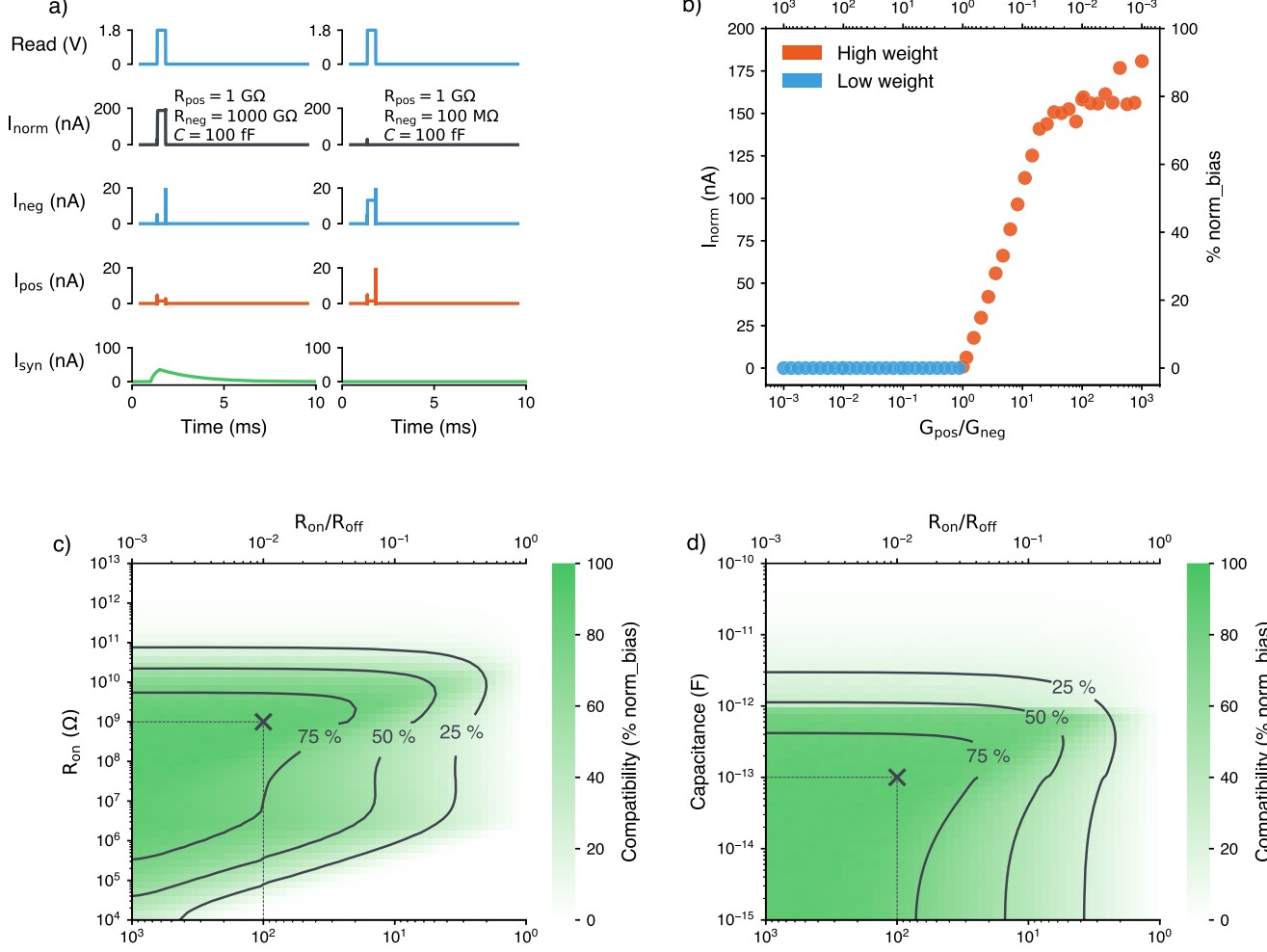

**Fig. 4 | Spectre post-layout simulations of the read protocol for the differential normalizer synapse on TEXEL. a** A read pulse with a width of 500 µs activates the normalizer circuit, sourcing $I_{neg}$ and $I_{pos}$. The circuit outputs a non-zero current, $I_{norm}$, if $I_{pos} > I_{neg}$, which is integrated by a DPI synapse, resulting in a current $I_{syn}$ sent to the neuron. The left panel shows high weight storage ($R_{pos} < R_{neg}$), eliciting a response, while the right panel shows low weight storage ($R_{neg} < R_{pos}$), where no current is integrated. **b** With $R_{pos} = 1\,G\Omega$, device capacitance of $C = 100\,fF$, and a read pulse width of 500 µs, the relative resistances of both devices are varied by

sweeping $R_{neg}$. The average output current of the normalizer circuit is measured as a % of norm_bias, showing non-zero current when the positive device's conductance exceeds that of the negative device. **c** Simulations explore device characteristics' impact on compatibility with TEXEL. The cross ( × ) represents a device with $C = 100\,fF$, $G_{on}/G_{off} = 100$, $R_{on} = 1\,G\Omega$, and a read pulse width of 500 µs. Heatmaps indicate average current from the normalizer as a percentage of norm_bias. **d** A sweep of the device's capacitance versus its on/off ratio is shown with $R_{on}$ fixed at 1 GΩ.

varying the on-off conductivity ratio and capacitance of the device (Fig. 4d), here the same metric of compatibility is used. This is an additional memristive device constraint that must be satisfied to ensure successful integration with CMOS and one that is often overlooked. Table 1 presents the integration specifications derived

from the aforementioned simulations, operating voltages and circuit footprints.

To more comprehensively evaluate the operation of the differential normalizer synapse, we conducted Spectre simulations using an open-source VCM compact memristive device model[52]. The simulation

protocol involved repeatedly switching a single synapse between potentiated and depressed states, corresponding to high and low conductivity states, respectively (Fig. 5a). During each cycle, a presynaptic spike was sent to the synapse, and the resulting postsynaptic current was measured through the differential normalizer circuit. Crucially, the postsynaptic current was only significant when the synapse was in the potentiated state. The simulation was run over several minutes, encompassing numerous device cycles to mimic the weight dynamics of an SNN during learning and synaptic consolidation. The compact model incorporated cycle-to-cycle variability, which is evident in the histogram of charge sourced by the postsynaptic current (Fig. 5b). Despite the inherent variability in conductivity states between cycles, the proposed circuitry effectively discriminates between high and low weight states, with no overlap in the encoded states.

## Power measurements

Extensive power measurements were conducted on the TEXEL chip using a femtoampere Source Measurement Unit (SMU) to assess its power distribution across operations for the analog and digital power sources. These measurements were performed on the CMOS chip and do not account for integrated devices. When considering a complete system with integrated memristive devices, the total power consumption would include additional contributions from the switching and reading operations of these devices. Memristive technologies that require low switching energy and can operate with low read voltages and currents would be optimal for maintaining energy efficiency in a complete system, particularly for learning operations that involve frequent weight updates. Figure 6a, c show how the dynamic power consumption varies with the global spike rate of the chip, this was modulated by increasing the DC input bias for all neurons. The total

dynamic power consumption is divided into the contributions of the isolated digital, analog and padframe power supplies. The digital circuitry accounts for all programming and spike routing, while the analog circuitry encompasses neurons, synapses, learning circuits, and analog parameter generation. During these experiments, the learning circuitry was turned off. The energy per spike was also calculated for varying spiking rates (Fig. 6b, d). Figure 6e, f show the same power contribution breakdown for synaptic operations with the energy required per operation. This experiment was performed by increasing the input spike rate over the Address Event Representation (AER) bus, randomly addressing all synapses on the chip, over both cores. Figure 6g shows the breakdown of the static power consumption of the chip, measured at 27.4 μW.

## On-chip spiking neural network

We evaluated the network-level functionality and on-chip learning capabilities of the TEXEL processor by implementing an abstract set-membership task within an SNN, using the formalism of VSAs, also known as Hyperdimensional Computing[53–56]. VSAs represent symbolic data with high-dimensional random vectors, known as *hypervectors*, such that the information is distributed across many neurons. They are particularly well-suited for neuromorphic hardware implementation, in part due to their emergent robustness properties at high dimensionality, making them naturally compatible with the circuit-to-circuit variability in mixed-signal computing substrates[57–60]. Specifically, we generated High Dimensional (HD) vectors to represent semantic objects (vehicles and colors), then bundled them together by a superposition operation to generate HD vectors representing sets of objects (Fig. 7a). We used sparse binary block code vector representations[61,62] such that vectors were partitioned into blocks of length $L$, with each block containing a single randomly positioned 1. Consequently, each atomic hypervector $\mathbf{v} \in \{0, 1\}^N$ ($N = 2000$) maintained a sparsity of $1 - \frac{1}{L}$. The bundled set hypervectors are given by

$$\mathbf{v}_{vehicles} = \mathbf{v}_{car} \oplus \mathbf{v}_{helicopter} \oplus \mathbf{v}_{bike} \tag{1}$$

$$\mathbf{v}_{colours} = \mathbf{v}_{magenta} \oplus \mathbf{v}_{cyan} \oplus \mathbf{v}_{yellow} \tag{2}$$

where $\oplus$ is an element-wise add-and-threshold operation, implementing the bundling, and the vectors on the RHS are independently generated. These bundled representations can then be probed to verify whether an object exists within the sets they represent. Set membership is determined by measuring the overlap between a query

### Table 1 | TEXEL memristive device compatibility requirements

| | Read Voltage (V) | Set Voltage (V) | Reset Voltage (V) | Area (μm²) |
|---|---|---|---|---|
| Min. | 0 | − 5 | − 5 | – |
| Max. | 5 | 5 | 5 | 114 |
| | Total Capacitance (pF) | Capacitance/Area (F/cm²) | $R_{on}$ (GΩ) | $G_{on}/G_{off}$ – |
| Min. | – | – | – | 10 |
| Max. | 10 | $8.8 \times 10^{-7}$ | 10 | – |

Entries in the second row are derived from post-layout simulations (Fig. 4), 50% is taken as a confidence threshold for compatibility.

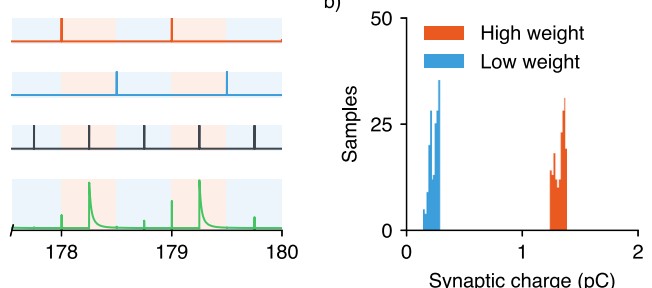

**Fig. 5 | Spectre simulation of the differential normalizer synapse integrated with a VCM device compact model. a** Potentiation (POT) and depression (DEP) events arriving at the synapse execute the complementary set and reset of the two devices in the differential configuration. Presynaptic spikes arriving at the synapse read the state of the devices through the normalizer circuit and the rescaled current is integrated by a DPI circuit providing an EPSC, $I_{syn}$, to the associated neuron. Sequential

POT and DEP events cycle the weight stored by the synapse, with presynaptic spikes occurring between cycles. **b** During the cycling of the state of the differential normalizer synapse, the current sourced through the DPI synapse circuit during a presynaptic spike was integrated to calculate the synaptic charge conveyed to the neuron. The distribution of this charge for high and low weight storage is shown, attributed to the cycle-to-cycle variability simulated in the compact model of the memristive device.

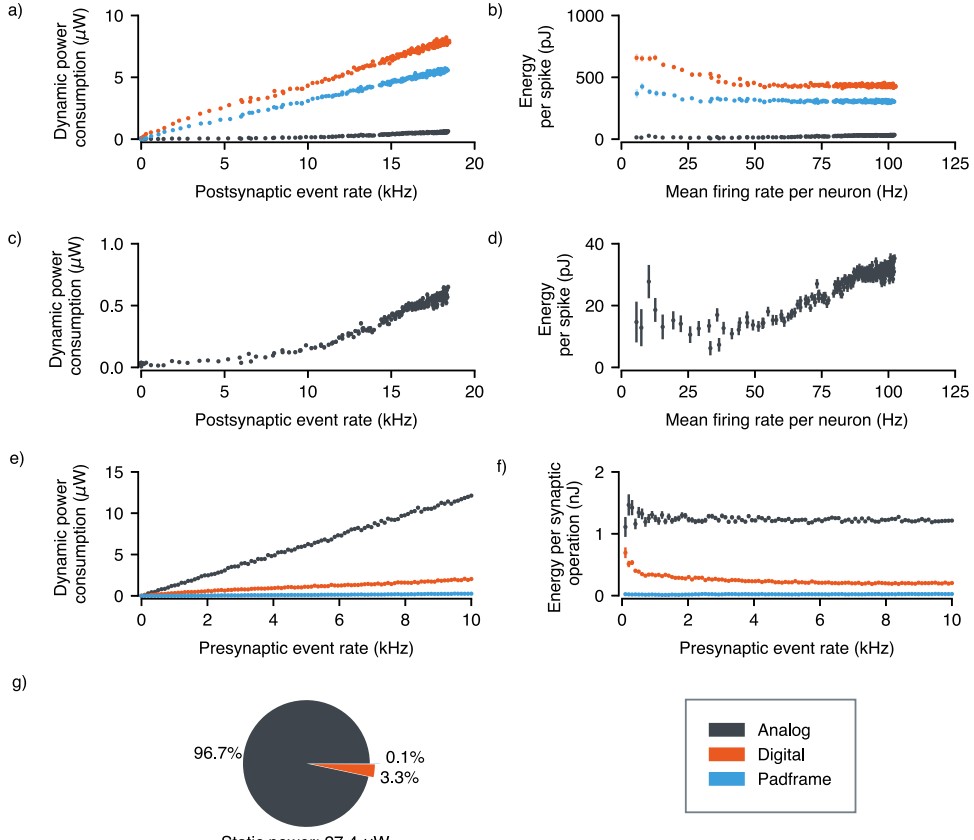

**Fig. 6 | Dynamic and static power measurements of the TEXEL chip, including energy consumption for synaptic operations and neuron spikes. a** Dynamic power consumption versus postsynaptic event rate, measured for the three isolated power supplies. **b** Energy per spike for increasing mean firing rates across each power supply. **c** Dynamic power consumption of the analog power supply against postsynaptic event rate. **d** Energy consumed per spike versus mean firing rate per neuron, for the analog power supply. **e** Dynamic power consumption during random synaptic stimulation at increasing input event rates. **f** Energy consumption per synaptic operation against input event rate. **g** Breakdown of static power consumption while neurons are inactive and synapses are unstimulated. All error bars represent measurement uncertainty.

vector and the bundled vector stored in memory, with significant overlap indicating membership.

This operation can be efficiently realized by a single-layer neural network classifier. Due to the on-chip network size constraints, we realized the 2000-dimensional vector representations using 80 neurons randomly distributed across the chip, with two populations of 40 neurons (each utilizing 50 synapses) encoding each class (vehicles or colors) (Fig. 7b). Initially, all plastic synaptic weights were configured to the depressed state (Fig. 7d). The network was then presented with spiking versions of $\mathbf{v}_{vehicles}$ or $\mathbf{v}_{colours}$, with teaching signals provided through non-plastic synapses (Fig. 7c). The temporal relationship between presynaptic (pattern) spikes and postsynaptic spikes (induced by presynaptic teacher spikes) proved sufficient to potentiate a subset of synapses for pattern learning (Fig. 7e). Following this one-shot learning phase, plasticity was switched off to prevent further learning, and test patterns representing objects stored within the learned sets were presented (Fig. 7f, g). When subsequently presented with any of the atomic hypervectors (e.g., $\mathbf{v}_{cyan}$), the neurons designated to represent each class gave the correct responses (Fig. 7h, i). Importantly, when presented with an object not stored in any set, the network showed no response, demonstrating successful specification to target HD representations. The implementation of these VSA operations through Hebbian auto-associative memories in neuromorphic hardware represents a promising step toward more sophisticated semantic and analogical reasoning in ultra-low-power computing systems[63].

## Discussion

Recent advancements have produced only a few successfully co-integrated large-scale CMOS-memristor neuromorphic systems[64–68], with most relying on foundry assistance[65–68]. This lack of co-integration is a key challenge in advancing such systems, emphasizing the importance of wafer-level integration platforms to establish compatibility with CMOS technology and to progress neuromorphic chip development. In this work, we introduced TEXEL, an SNN processor with on-chip learning circuits capable of interfacing with a large range of memristive devices and their operation requirements (Table 1). TEXEL functions in both full-CMOS and device-integrated modes, offering versatility to explore emerging memristive technologies within the context of a spiking neuromorphic system. The processor supports a wide range of device interfacing options, including read-write pulse widths from 10 ns to 100 ms, continuous read and precharge modes (see Supplementary Section A4), and high-voltage compatibility up to 5 V. The architecture further extends its compatibility to include both two-terminal devices and three-terminal technologies such as Ferroelectric Field-Effect Transistors (FeFET)[69,70].

The chip prioritizes flexibility over efficiency by supporting multiple device debug modes and allowing operation without devices for CMOS circuit verification. This design approach enables thorough benchmarking and debugging of CMOS-memristor circuits through comprehensive signal monitoring capabilities (see Supplementary Table 1). However, while TEXEL's broad compatibility with Non-Volatile Memory (NVM) devices provides versatility, it introduces significant area overhead (see Table 2 and Supplementary Fig. S4a). This overhead

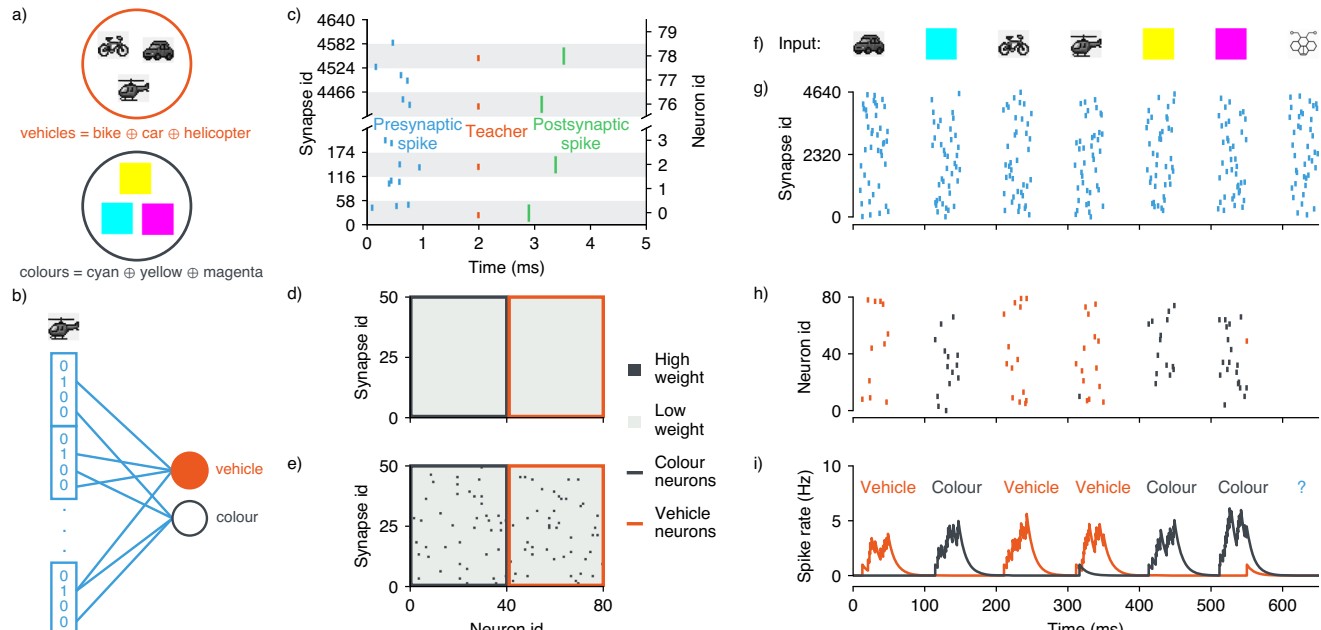

**Fig. 7 | SNN implemented on the TEXEL chip. a** Sparse HD vectors representing semantic objects were bundled ( ⊕ ) to form vectors representing sets of objects. **b** The single-layer network configuration, HD vectors were converted to spike trains determined by the presence of 1s in the vector, where each 1 was translated to a single spike sent to a synapse based on the mapping between vector dimension and synapse/neuron id. **c** Spike patterns representing sets were presented to the SNN along with teaching signals, provided through static excitatory synapses. The plastic synapses receiving the input spike pattern and the teacher signals potentiated their state using the learning circuitry. **d** A readout of the weights matrix of the SNN before learning the input patterns, all synapses are in the low weight state. **e** A readout after learning, a subset of synapses have potentiated to the high weight state. **f** During inference, the SNN is sent spike patterns (**g**) representing classes stored within the learned sets of colors and vehicles. **h** The response of the network to the input patterns, the neuron populations corresponding to the set in which the object is stored respond with postsynaptic spikes. **i** A readout of the spike rate of the populations encoding for the sets demonstrate the correct response to the input spike vectors. An object not contained in either learned set is shown and network does not provide a response.

primarily stems from necessary components such as voltage level shifters and device interfacing circuitry like the differential normalizer circuit. Since these circuits scale with the number of synapses, which dominate the chip's silicon real estate, their impact on area efficiency is substantial. Nevertheless, as NVM technology advances toward higher numbers of distinguishable states, a more favorable trade-off between area requirements and synaptic resolution is anticipated. Once a specific NVM technology is selected, future iterations should aim to optimize both density and performance by eliminating the redundant circuitry currently needed to accommodate multiple device types.

The integration of memristive devices and materials with CMOS neuromorphic systems extends beyond storing and reading synaptic weights. These technologies can be incorporated into neuron circuits[71] and learning mechanisms[72], enhancing characteristics such as time constants and dynamic behaviours. Additionally, emerging devices and materials have shown significant promise in sensory applications[73,74], rendering them particularly appealing for integration into sensory front ends that can be interfaced with always-on neuromorphic chips. Research has also revealed how beyond-CMOS devices can enable advanced network features, including the realization of synaptic delays[34] and small-world network topologies[75], further enhancing functional capabilities of neuromorphic systems. Complementing these advances, ongoing development of monolithic 3D structures presents a promising direction to increase device density and reduce interconnect lengths, addressing fundamental challenges in scaling neuromorphic architectures[76–78]. Collectively, these capabilities position memristive device technology as a key component in the development of efficient and adaptable electronic architectures. Within this evolving landscape, TEXEL serves as a platform aiding the realization of CMOS-memristor neuromorphic systems that can scale effectively while leveraging the advantages of emerging device technologies.

## Methods

### Chip architecture

With 2 cores of 90 neuron blocks, TEXEL hosts 180 neurons each with 58 complex synapses (Fig. 1b). The chip's digital periphery operates asynchronously, utilizing handshake protocols between functional blocks[79]. Robustness was tested through extensive testing for variable switching delays, eliminating the reliance on specific timing constraints. Spike I/O and register operations share an asynchronous pipeline tailored for AER. Demux circuits route incoming packets to either the spike decoder or register block. The decoder translates external AER spike packets, while the encoder processes on-chip neuron spikes for transmission off-chip. The register block comprises 64 23-bit asynchronous memory arrays (per core) used for biasing and programming, each capable of parallel read or write operations. All analog circuitry is biased using a 12-bit DAC (Digital-to-Analog Converter) (see Supplementary Section A3). To enable the integration of two- and three-terminal NVM devices there is interfacing circuitry including terminal contacts placed within each plastic synapse in every neuron block[1,80,81] (see Supplementary Fig. S4a). Figure 1 shows the embedding of the neuron blocks and synapses within the chip architecture.

### Neuron circuits

The Adaptive Exponential Leaky Integrate-and-Fire (AdExLIF) neuron circuit integrated on TEXEL is the latest iteration of a continuing design evolution that has undergone multiple enhancements to optimize performance[1,6,24,82–84] (see Supplementary Fig. S2). The implementation of the neuron draws inspiration from the improvements detailed in[25], focusing on minimizing power consumption and reducing mismatch. The neuron dynamics are driven by two inputs: a DC input and a somatic input current from the synaptic fan-in, enabling network-level experiments. The somatic input DPI models the neuron's leak

**Table 2 | Comparison of TEXEL with other silicon-verified memristor-SNN chips**

| Chip | TEXEL [this work] | ISSCC'20[65] | ISCAS'23[67] | NeuRRAM[66] |
|---|---|---|---|---|
| Design | mixed-signal | mixed-signal | mixed-signal | mixed-signal |
| CMOS technology | 180 nm | 130 nm | 130 nm | 130 nm |
| Device type | any BEOL current based | RRAM | OxRAM | RRAM |
| Device terminals | 2–3 | 2 | 2 | 2 |
| Number of devices | 19 k | 65 k | 4 k | 3.14 M |
| Area including I/O | 67.5 mm² | – | - | 158.76 mm² |
| Core area | 44.98 mm² | 1.79 mm² | 0.180 mm² | – |
| Neuron model | AdExLIF | IF | IF | IF |
| Number of neurons | 180 | 256 | 64 | 12 k |
| Number of synapses | 10 k | 65 k | 4 k | 3.14 M |
| Full parallel write | yes | – | column–wise | – |
| In-memory plasticity* | ✓ | ✗ | ✗ | ✗ |
| Learning rule | STDP & SDSP | – | S-STDP | – |
| Energy/spike NeuOp | 25.9 pJ @ 80 Hz | 0.0139 pJ/MAC | – | 0.121 pJ |
| Chip | NElec'18[40] | IEDM'19[68] | VLSIT'19[64] | |
| Design | memristor | mixed-signal | mixed-signal | |
| CMOS technology | – | 130 nm | 150 nm | |
| Device type | HfO$_x$ RRAM | OxRAM | HfO$_x$ RRAM | |
| Device terminals | 2 | 2 | 2 | |
| Number of devices | 74 | 13.5 k | 64 k | |
| Area including I/O | 0.56 mm² | – | – | |
| Core area | – | – | – | |
| Neuron model | stochastic LIF | IF | IF | |
| Number of neurons | 8 | 10 | 256 | |
| Number of synapses | 64 | 1440 | 65 k | |
| Full parallel write | yes | – | – | |
| In-memory plasticity* | ✓ | ✗ | ✗ | |
| Learning rule | Hebbian LTP | – | – | |
| Energy per spike/NeuOp | – | – | 0.257 pJ/MAC | |

*The plasticity rule is implemented in-memory with local circuits.

conductance, integrating synaptic currents into the membrane capacitance, producing a membrane current representing the neuron state variable. Between the somatic DPI and spike generation, three modules control membrane current dynamics: a threshold, exponential and refractory module. The threshold module, implemented with a low-power current comparator, triggers a spike at the moment the membrane current exceeds the spiking threshold. The exponential module, implemented with a current-based positive feedback, accelerates the membrane current increase when it is closer to the spiking threshold. Once the neuron generates a spike, the refractory module keeps the neuron silent for a certain time set by the refractory period bias. Furthermore, there is an adaptation module, implemented with a pulse extender and a negative-feedback low-pass filter circuit. This is activated with each output spike event, integrating the neuron's recent spiking activity. All aforementioned modules can be controlled using seven tunable biases. The neuron circuit is designed to be compatible with AER circuits therefore an asynchronous digital handshaking block is incorporated to transmit spikes as address-events through the AER pipeline.

**Synaptic circuits**
Each neuron on the TEXEL chip has a synaptic fan-in of 58 synapses, 54 plastic and 4 non-plastic (static). Non-plastic synapses are realized through DPI circuits and activate in response to a presynaptic spike, producing a current with an amplitude that is tunable. Consequently, they can be deactivated by setting the weight bias current to zero. The nature of the non-plastic synapses is predetermined, with two per-

neuron designated as excitatory and two as inhibitory (Fig. 1b). The weight of the plastic synapses, updated according to the on-chip local learning rule, is stored on a capacitor on a short-time scale and discretized into two stable states on a long-time scale. The weight update occurs in the analog domain, while the long-term storage takes place in the digital domain. The nature of the plastic synapses (excitatory or inhibitory) can be configured on-chip. Excitatory synapses inject a positive current into the soma, while inhibitory ones draw current away from it. The total synaptic activity, computed as the sum of weighted currents, is transmitted to four different DPIs, each independently tunable.

**Learning circuits**
Within each plastic synapse there exists a CMOS implementation of the BiCaLL rule[44] that can be enabled, making use of signals local to each synapse to facilitate either Hebbian or anti-Hebbian Spike-Driven Synaptic Plasticity (SDSP) (Fig. 3e). A pre-trace, realized by a DPI circuit[51], maintains a decaying memory of the presynaptic spike train. If this trace exists between an upper and lower threshold then with the co-occurrence of postsynaptic spikes the synaptic weight is depressed. In parallel, depression can also occur if the post-trace, a short term memory of postsynaptic activity, is above a low threshold and a presynaptic spike occurs. Potentiation occurs on a postsynaptic spike during which the value of the presynaptic trace is sampled from such that the magnitude of potentiation is proportional to the presynaptic trace at that time[30]. A smooth third trace, realized by a Second-order Differential Pair Integrator (SoDPI) circuit[85], is used to track the

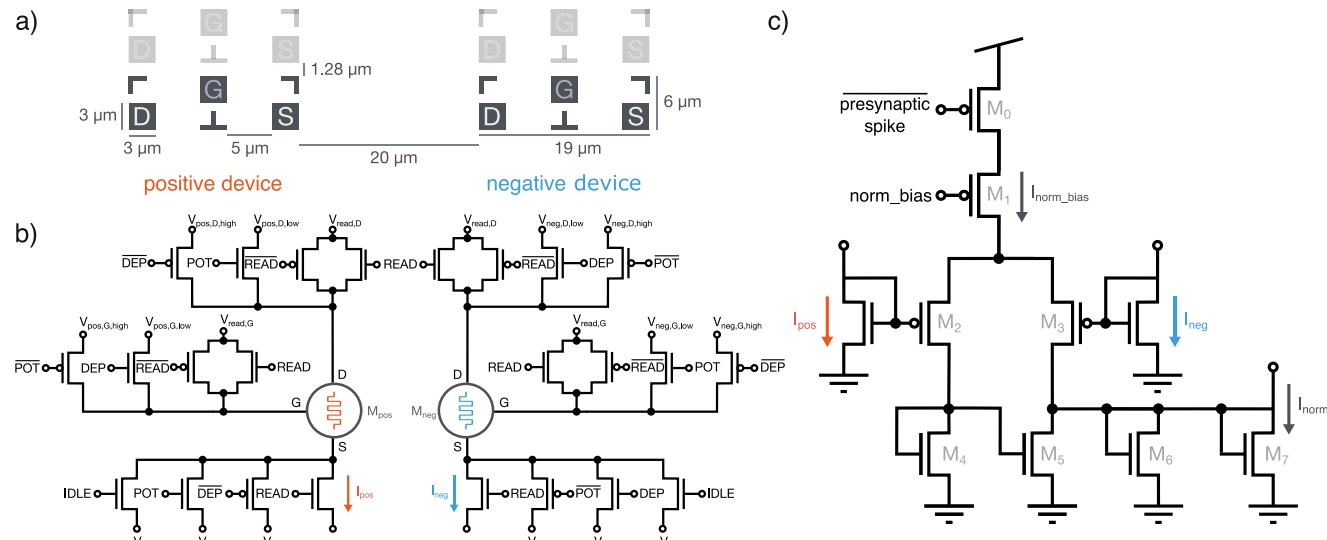

**Fig. 8 | The footprint and schematic of the per-synapse device interface terminals and schematic of the differential normalizer circuit. a** A diagram illustrating the physical dimensions and spatial arrangement of the source, drain, and gate contacts for two- or three-terminal devices. Each synapse deploys two devices configured differentially, serving as both positive and negative components. The diagram also provides information on the spacing between synaptic rows, depicting the distances between adjacent devices in each synapse. **b** Schematic of device interface circuitry. All voltages can be set in the range 0 V to 5 V in order to read or write both devices in the differential configuration. **c** The differential normalizer circuit functions to compare the currents generated by positive and negative devices during a device read, prompted by a presynaptic spike. It evaluates the disparity between these currents and generates an output current, denoted as $I_{norm}$, which is proportional to the normalized discrepancy between $I_{pos}$ and $I_{neg}$. Moreover, $I_{norm}$ is exclusively non-zero when $I_{pos}$ surpasses $I_{neg}$ and can be modulated by the bias norm_bias. Consequently, the output represents the binary state of the synapse, and the sourced current is directed towards a DPI circuit for further processing.

neurons' activity, representing the postsynaptic neuron's $Ca^{2+}$ concentration. The upper and lower thresholds of the $Ca^{2+}$ trace establish a stop-learning region, restricting synaptic plasticity to occur only within this range. The weight is stored as a voltage as shown in Figure 3a and is discretized via a voltage threshold. Additionally, a bistability circuit is employed such that over the long time scale the weight drifts towards a binary value. The temporal dynamics of the aforementioned traces, strength of the potentiation/depression events, bistability slew rates and thresholds can all be varied through the biasing of the analog circuitry.

**Memristive device integration**
To support large-scale integration of plastic memristor-based synapses, the chip is designed with a device-agnostic architecture, ensuring high flexibility and offering multiple probing configurations for different memristive devices. This design accommodates both two- and three-terminal devices, supporting a broad range of operating voltages and currents (see Table 1). Device behaviour can be monitored either through on-chip read-outs of output currents during operation or via off-chip access to all device terminals through the interface circuit (see Supplementary Table 1). Full access to the device terminals enables external burn-in or programming of the memristive devices.

To facilitate BEOL integration, each terminal is accessible through a high-level metal contact with spacing and sizing depicted in Fig. 8a. Three branches in the interface circuitry employ n- and p-type transistors, along with transmission gates, to deliver voltage pulses for reading device states or for potentiating or depressing synaptic weights (Fig. 8b). The operation voltages are provided off-chip as inputs to the padframe with a maximum voltage of 5 V. Digital signals to the transistor gates are internally controlled by a synapse controller circuit which implements synaptic operations and weight updates. We note that an extra idle transistor and idle signal is used to facilitate the possibility of pre-charging the device between read pulses and allow a better distinction between their HRS and LRS currents (see Supplementary Section A4).

Often, device operation specifications are not immediately compatible with the technology node and cannot be compensated for by voltage scaling or pulse length modulation. This can occur when currents are too low or too high, device variability is significant, or the resulting output ranges are undefined. In these cases, scaling and normalizing circuits can be employed. Given this initial assumption about the properties of a device aiming for compatibility, the TEXEL chip uses the difference in state of two devices to store the synaptic weight of each plastic synapse. Therefore the canonical on-chip operation protocol for memristive devices is binary and complementary. As a result, while using the on-chip plasticity, devices are only switched in a binary operation between HRS and LRS, and always in a complementary fashion where if one is in its HRS, the other will be in its LRS. A differential normalizer circuit is used to compare the responses of two devices[38] when the synaptic weight it being read. When the synapse is addressed for a read, at a presynaptic spike, the currents are sourced from the devices, $I_{pos}$ and $I_{neg}$, and the normalizer circuit (Fig. 8c) rescales and rectifies the detected difference to the output current range required by the DPI synapse, $I_{norm}$. The rescaling factor of the output current can be modulated by the bias norm_bias.

Since many memristive devices use the same terminals for both reading and writing, they require exclusive control to prevent conflicts. In other words, when a read and write instruction occur simultaneously, a decision must be made regarding which operation to execute first. To manage this, each plastic synapse has a dedicated control circuit that ensures mutual exclusivity between read and write pulses (see Supplementary Fig. S3). Read instructions are prioritized, therefore if both commands occur concurrently, the write pulse is applied only after the read operation is completed.

**Memristive device simulations**
Spectre simulations were performed using the circuitry illustrated in Fig. 8, employing a compact VCM device model implemented in Verilog-A[52]. The voltages applied to the transistors for potentiating, depressing, and reading the device were configured within the operating ranges of the chip (Table 1) and maintained compatibility with

the device model. A norm_bias current of 200 nA was used during the simulation. The synapse was repeatedly cycled between high and low conductivity states by applying depression and potentiation signals, with complementary device switching occurring through the device controller logic. During each cycle, a presynaptic spike was delivered to the synapse to induce a postsynaptic current. This current was measured and compared to the known state of the synapse to verify that the correct current magnitude was sourced relative to the conductivity state of the device. The simulation was run for a total of 3 minutes, with the synapse being cycled at a rate of 1 cycle per second and read operations performed at 2 presynaptic spikes (reads) per second.

**On-chip spiking neural network**

VSA hypervectors were generated off-chip with a dimensionality of 2000, using sparse binary block codes. Each vector was partitioned into blocks of length $L = 20$, with each block containing a single randomly positioned 1, maintaining a sparsity of 95%. Object representations were bundled ($\oplus$) off-chip using element-wise addition and thresholding at 1 to create set vectors for vehicles and colors. The 2000-dimensional hypervectors were mapped to spike trains using a one-hot single spike encoding scheme, where 0 in the vector corresponded to no spike and 1 corresponded to a single spike on the associated synapse. These inputs were distributed randomly across 80 neurons on a single core of the TEXEL processor, with synaptic fan-in connections spanning the chip. Two distinct populations of 40 neurons each were designated to encode the two sets (vehicles and colours). Prior to learning, the weights of all plastic synapses were initialized to the low weight state. During the learning phase, each set pattern was presented simultaneously to both neuron populations, with the target population receiving additional teacher signals via non-plastic synapses. These teacher signals induced postsynaptic spikes in the designated population. The temporal coincidence between presynaptic spikes (representing the input pattern) and the teacher-induced postsynaptic spikes triggered synaptic strengthening according to the STDP relationship shown in Fig. 7c. This one-shot learning procedure was performed sequentially for both populations to encode their respective set vectors (vehicles and colours). For inference, vectors representing individual items from each set were presented simultaneously to both populations, and neuronal activity was recorded. Membership was determined by the response of the respective neuron population, with significant activity indicating set membership. The synaptic weight matrix was read out before and after the learning phase to verify changes in connectivity strength.

## Data availability

All data and methods needed to evaluate our conclusions are presented in the main text and Supplementary Material. No extensive datasets were generated as a result of this study.

## Code availability

The code generated to interface with the chip presented in this study is available at https://github.com/async-ic/uC-chip-interface-arduino[86].

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

## Acknowledgements

The authors would like to thank Adrian Whatley, Vincent Jassies and Herman Adema for their technical support in developing PCBs, µC firmware and the TEXEL API. Additionally we would like to thank Nicoletta Risi and Matei Zainea for their investigations into algorithms and hardware. Thanks to Ton Juny Pina for his help soldering PCBs. Finally, we would like to thank Erika Covi, Luca Fehlings, Paolo Gibertini, Giuseppe Leo and Ton Juny Pina for their feedback on the manuscript. This work was supported by: the EU H2020 RIA project BeFerroSynaptic (871737) (A. R., E. C., G. I., J. C., M. F., P. K., and L. B. L.), the EU H2020 MSCA projects NeuTouch (813713) (E. C. and M. M.) and MANIC (861153) (E. C., W. S. G.), the ERC Synergy grant SWIMS (101119062) (E. C.), the Deutsche Forschungsgemeinschaft (DFG, German Research Foundation) projects MemTDE (441959088, part of the DFG priority program SPP 2262 MemrisTec, 422738993) (E. C., H. G.) and NMVAC (432009531) (E. C., M. C., and M. Z.). The authors would like to acknowledge the financial support of the CogniGron research center and the Ubbo Emmius Funds (University of Groningen) (E. C., H. G., M. C., M. F., M. M., O. R., P. K., and W. S. G.). The authors would like to thank IC Manage, Inc for providing us with their Global Design Platform XL for design data management.

## Author contributions

Conceptualization - A. R., E. C., G. I., H. G., J. C., M. C., M. F., M. M., O. R., P. K., and W. S. G.; Methodology - A. R., E. C., G. I., H. G., J. C., M. C., M. F., M. M., O. R., P. K., and W. S. G.; Software/Hardware - A. R., E. C., G. I., H. G., J. C., M. C., M. F., M. M., O. R., P. K., and W. S. G.; Investigation - A. R., E. C., G. I., H. G., M. M., and O. R.; Writing - A. R., E. C., G. I., H. G., M. C., M. F., L. B. L., M. M., M. Z., O. R., and P. K.; Visualization - E. C., G. I., H. G., M. C., M. M., and O. R.; Supervision - E. C. and G. I. The following authors contributed significantly to the CMOS design of the TEXEL chip - A. R., E. C., G. I., H. G., J. C., M. C., M. F., M. M., O. R., P. K., and W. S. G. Authors O. R. and M. M. conducted the work while affiliated with the BICS lab and CogniGron at the University of Groningen.

## Competing interests

The authors declare no competing interests.
