## [Transparent Peer Review file · Nature Communications]

A neuromorphic processor with on-chip learning for beyond-CMOS device integration

Corresponding Author: Mr Hugh Greatorex

Version 0:

Reviewer comments:

Reviewer #2

(Remarks to the Author)

This paper proposes TEXEL, a versatile research platform for large-scale BEOL device integration in neuromorphic systems with on-chip learning capability. The platform may contribute to the efforts of advancing the efficient development of large-scale BEOL device integration in neuromorphic systems as a flexible tool, provided that this tool is well verified. However, the platform has not yet been verified by the integration of BEOL devices such as memristors or FeFETs, in particular, in a large scale, making the feasibility of the approach arguable. I would thus recommend this work to be further considered only if the authors can address the following concerns.

- The authors claim that a key strength of the proposed platform is its capability to support large-scale BEOL integration of various devices within the context of an SNN processor. However, it has been agreed by most researchers that the main challenge in such integrated large-scale memristor+CMOS neuromorphic systems lies in the fabrication of the large-scale memristor arrays instead of the underlying CMOS. Representative issues include the thermal budget constraints and the degradation of device characteristics in large-scale fabrication. This makes the readers unclear regarding the challenges and contributions of the proposed CMOS-based neuromorphic platform.

- The novelty of the CMOS circuits in the proposed platform is also unclear. The neural circuits seem to be referred to existing works, and the improvements made to the synaptic and learning circuits are not explicitly clarified or highlighted in the paper.

- A major limitation of this work is the lack of demonstrated large-scale BEOL device fabrication on the proposed platform, which is essential for the verification of the concept or the potential contribution. Despite the presented analysis through simulations, the devices are simply modeled using only a few high-level parameters, i.e. resistance, capacitance and on/off ratio without experimental measurements of fabricated devices or the use of compact models, making the feasibility of the proposed platform very questionable.

- How does the BEOL device non-idealities such as variations and drifts affect the system performance? An explanation on the reliability of the proposed chip would be appreciated.

- The authors also highlight in-memory plasticity as another strength of the proposed platform, but the benefit of achieving plasticity is not shown in the results. As shown in the energy efficiency comparison, the proposed platform does not appear to outperform the existing works.

- Is the energy consumption of the resistor devices included in the energy evaluation? If so, how does the resistor device modeled and evaluated? The authors should present the setups with more details.

- What is the benefit of supporting both STDP and SDSP learning rules?

- The proposed platform is claimed to be an SNN processor, but its functionality is not demonstrated in SNN applications. At least one SNN application implemented on the proposed chip should be presented with accuracy or other metrics.

- Figure 3b and 3c are confusing for readers. What is the difference in the setups for the two figures? More explanations in

the caption or additional legends would be appreciated.

Reviewer #3

(Remarks to the Author)

This work presented by Greatorax et al. proposed to demonstrate a mixed-signal neuromorphic architecture — TEXEL and tried to develop the integration of on-chip learning circuits and novel two- and three-terminal devices by using this system. The construction of neuromorphic chips is a hot and interesting topic. The authors have demonstrated a complete and detailed circuit system to implement multiple synaptic plasticity functions. Nevertheless, the authors lack detailed clarification on material and device selection based on this system. In addition, the lower energy operation still needs further investigation. Thus, I cannot recommend the paper to be published in the journal of Nature Communications in its current form. The comments are shown as follows:

1. Although a large number of synaptic plasticity functions such as STDP, SRDP and LIF can be fully reproduced in the circuit system, the above plasticity is only a single synaptic or neuronal function in the biological system, and can be achieved on a single analog memristor /RRAM cell. What are the advantages of achieving plasticity through this circuit system compared to a single memory device? Can more complex SRDP functions such as triplet STDP and BCM learning rules, be implemented by this more complex circuit chip?
2. The authors claim that two - or three-terminal devices can be integrated based on the system for on-chip learning. The authors need to clarify which memory devices have more energy-efficient learning based on this system. Based on this circuit system, the authors also need to give a detailed list of memristive materials to obtain the best performance.
3. It is well known that the energy consumption of a synapse to complete a synaptic event is only 1-10fJ. However, the energy consumption provided by the author in Table 1 is 25.9PJ, which is much higher than the energy consumption of biological systems and other chip systems. Therefore, it is necessary for the authors to improve the existing energy consumption of the circuit.

Version 1:

Reviewer comments:

Reviewer #2

(Remarks to the Author)

The authors have answered most questions. However, the advancement of this work is still doubtful. The main limitation is that the demonstration of both CMOS circuits and BEOL NVM devices is not meeting the expectation of sufficient completeness. There have been several published M3D works [1][2] that have complete demonstration of CMOS circuits and other BEOL devices such RRAM, CNT, etc. Although these works do not focus on SNN applications, they are still more advanced than the CMOS-only demonstration. Considering the reported energy efficiency of this work is not outstanding, I suggest the authors submit to other venues other than Nature Communications.

[1] Y. Li et al., "Monolithic three-dimensional integration of RRAM-based hybrid memory architecture for one-shot learning," Nat. Commun., vol. 14, no. 1, p. 7140, 2023.

[2] Y. Zhang et al., "Monolithic 3D Integration of Multi-Layer CNT-CMOS/RRAM Macros for Mixed-Precision Analog-Digital Computing-in-Memory Architecture," in 2024 IEEE International Electron Devices Meeting (IEDM), 2024, pp. 1–4.

Reviewer #3

(Remarks to the Author)

The authors have addressed my major concerns. In my opinion, it may be published.

RESPONSE TO REVIEWERS FOR MANUSCRIPT:

TEXEL: A NEUROMORPHIC PROCESSOR WITH ON-CHIP LEARNING FOR BEYOND-CMOS DEVICE INTEGRATION

Hugh Greatorex, Ole Richter, Michele Mastella, Madison Cotteret, Philipp Klein, Maxime Fabre, Arianna Rubino, Willian Soares Girão, Junren Chen, Martin Ziegler, Laura Bégon-Lours, Giacomo Indiveri, Elisabetta Chicca

We are very grateful to all reviewers for the time and considerable effort spent on reviewing this manuscript, and we highly appreciate the quality and insightfulness of the comments. In the manuscript, new or significantly revised text is marked in blue. Below is a detailed account of the changes that have been made to address the reviewers' comments.

Replies to reviewer 2

Comment 2.1

The authors claim that a key strength of the proposed platform is its capability to support large-scale BEOL integration of various devices within the context of an SNN processor. However, it has been agreed by most researchers that the main challenge in such integrated large-scale CMOS-memristor neuromorphic systems lies in the fabrication of the large-scale memristor arrays instead of the underlying CMOS. Representative issues include the thermal budget constraints and the degradation of device characteristics in large-scale fabrication. This makes the readers unclear regarding the challenges and contributions of the proposed CMOS-based neuromorphic platform.

Reply: We sincerely thank the reviewer for their insightful comment regarding the challenges of large-scale BEOL integration of memristive devices. We fully agree with the reviewer's assessment that one of the primary challenges in CMOS-memristor neuromorphic systems lies in the fabrication of large-scale memristor arrays rather than in the underlying CMOS circuitry. These challenges, including thermal budget constraints and device characteristic degradation during large-scale fabrication, are indeed significant hurdles in the field.

We appreciate this opportunity to clarify that no effective BEOL integration has been demonstrated in the current work. The objective of this manuscript is to present how the CMOS circuits have been specifically designed to exploit BEOL devices for implementing a fully fabricated and silicon verified real-time spiking neural processor. To address the reviewer's concern, we have added a statement in the introduction section that explicitly acknowledges these challenges and better positions our contribution within this context.

The novelty of our work lies in the fact that TEXEL is, to the best of our knowledge, the first fully-fabricated CMOS device featuring a complex event-based analog/digital SNN computing substrate with on-chip learning capabilities specifically designed for integration with different types of memristive devices. This chip has been developed in collaboration with research groups specializing in memristive device fabrication, and specific, novel and deliberate design choices have been taken to facilitate the integration of research devices within and beyond the domain of these collaborations. In particular, the 3 branch device-agnostic device interface and differential normalizer circuits presented in Section 2.3 and Figure 9 allow for the accommodation of a wide range of device characteristics, as presented in Section 2.4. The pure fabrication and integration of said devices is a challenge beyond the scope of this paper but the novel solutions proposed, simulated and fabricated here will have a significant impact on the range of emerging memristive devices that can be accommodated. This positions TEXEL as an experimental platform to test and compare various constraints between different devices and large-scale fabrication methods.

For a possible integration of memristive layer stacks, lithography processes are also required to integrate the memristive devices into the contacts provided on the TEXEL chip. In Fig. R1 we show that this is possible with the chip designed here. In detail, the bottom electrode on which the memristive layer stack can be integrated is shown. The lithography process required for this was carried out on a standard mask-aligner, which is available in research clean rooms. While actual device integration is beyond the scope of this manuscript, work is currently underway with our collaborators to fabricate memristive devices on the chip. This ongoing effort directly addresses the feasibility concerns of realizing a fully integrated CMOS-memristor neuromorphic system. We hope that this manuscript will facilitate

collaborations with researchers developing BEOL-compatible processes, as the flexibility of the CMOS design has been specifically engineered with this collaborative research goal in mind. We thank the reviewer again for highlighting this important aspect, which has helped us improve the clarity and positioning of our work.

Fig. R1: Initial layer deposition process for the bottom electrode of memristive devices on the TEXEL chip. This evidences the first steps towards BEOL integration of memristive materials with the CMOS-based neuromorphic processor.

Comment 2.2

The novelty of the CMOS circuits in the proposed platform is also unclear. The neural circuits seem to be referred to existing works, and the improvements made to the synaptic and learning circuits are not explicitly clarified or highlighted in the paper.

Reply: We thank the reviewer for this comment and agree that highlighting the circuit advancements relative to previous work is important. To address this, we have added several clarifying sentences in Section 2. Our main contributions are two-fold: First, we have created a full-scale system by integrating state-of-the-art mixed-signal neuromorphic circuitry (combining subthreshold analog and digital asynchronous circuits) with memristive device interfaces. These circuits were specifically optimized for size and scalability, enabling the creation of macros suitable for fabricating a larger SNN processor. This represents a significant advancement from previous work, where circuits were limited to test configurations or small systems [1, 2]. Second, a major novelty, now emphasized in the revised text, is our implementation of a flexible device interface circuit alongside the controller and the layout for interfacing. These developments collectively enable a large-scale and versatile neuromorphic architecture than previously demonstrated.

Comment 2.3

A major limitation of this work is the lack of demonstrated large-scale BEOL device fabrication on the proposed platform, which is essential for the verification of the concept or the potential contribution. Despite the presented analysis through simulations, the devices are simply modelled using only a few high-level parameters, i.e. resistance, capacitance and on/off ratio without experimental measurements of fabricated devices or the use of compact models, making the feasibility of the proposed platform very questionable.

Reply: We thank the reviewer for this valuable comment that has significantly improved our manuscript. We fully acknowledge that demonstrating large-scale BEOL device fabrication on the proposed platform is crucial for concept verification and validating the potential contribution of this work.

As discussed in our response to the reviewer's first comment, we recognize the importance of conveying the feasibility of device integration on the TEXEL chip. To address this concern, we have included Fig. R1, which shows initial layer deposition for the bottom electrode of memristive devices currently being integrated on the chip.

We agree that our initial device modelling approach was simplified, using only a few high-level parameters (resistance, capacitance, and on/off ratio) to benchmark the compatibility ranges of the CMOS circuitry for device integration. While this approach provided useful compatibility ranges, it did not account for the transient dynamics of writing and reading real devices. To address this limitation, we have significantly expanded our validation methodology by:

1. Implementing additional simulations using a mature, open-source compact model that is compatible with our processor architecture. These more comprehensive simulations better capture the dynamic behaviour of memristive devices interfaced with our circuitry.
2. Adding new results to section 2.4 that presents these simulations and their outcomes, along with a corresponding methods section (see Section 4.6) describing the modelling approach in detail.

Furthermore, we would like to direct the reviewer to Supplementary Section A.4 which we have added to provide experimental information about memristive devices with fabrication methods and material stacks specifically designed to be compatible with the TEXEL processor. We believe these additions substantially strengthen the manuscript by providing more rigorous evidence for the feasibility of our approach while maintaining the focus on the novel CMOS architecture that forms the core contribution of this work. Thank you again for this comment that has helped us enhance the thoroughness and validity of our work.

Comment 2.4

How does the BEOL device non-idealities such as variations and drifts affect the system performance? An explanation on the reliability of the proposed chip would be appreciated.

Reply: We thank the reviewer for this important question. We have added a detailed paragraph to Section 2.3 that addresses how the differential normalizer circuit helps mitigate device variations and drifts. This discussion directly addresses your concerns about BEOL device non-idealities and system reliability. Specifically, we explain how the paired memristive devices in opposing states, when connected to the normalizer circuit, significantly reduce output current variability. We demonstrate that the configuration maintains robust performance as long as sufficient contrast between high and low resistance states is preserved. We refer to Figure 4c, which illustrates how device drift (which would appear as horizontal movement on the graph toward reduced $G_{\text{on}}/G_{\text{off}}$ ratios) affects performance, allowing for empirical evaluation of the circuit's tolerance to such variations. This analysis shows that the architecture remains effective even when facing the reliability challenges common to emerging resistive technologies.

Comment 2.5

The authors also highlight in-memory plasticity as another strength of the proposed platform, but the benefit of achieving plasticity is not shown in the results. As shown in the energy efficiency comparison, the proposed platform does not appear to outperform the existing works.

Reply: We thank the reviewer for this valuable point. This comment highlights the lack of demonstration of SNN applications that would showcase the benefits of our in-memory plasticity implementation. We addressed this omission by adding a new results (Section 2.6) and methods section (Section 4.7) that demonstrates the use of on-chip plasticity in network-level experiments. These additions provide concrete evidence of the functional advantages our plasticity mechanisms offer in practical neuromorphic computing applications. Regarding energy efficiency, we would like to emphasize that our neuron design utilises state-of-the-art subthreshold analog circuits presented in [2], optimized for the technology node used in the TEXEL chip. In particular, the TEXEL neuron circuit consumes only tens of picojoules per spike at biological firing rates (<100 Hz). This is an order of magnitude more efficient than a similar design using the same technology node [3], and comparable to a design implemented in more advanced nodes [4]. This energy efficiency reflects several key circuit-level optimizations, detailed in [2] and replicated in the TEXEL neuron design. Furthermore, when compared to a neuron design employing switched-capacitance techniques [5], the TEXEL neuron achieves a two-order-of-magnitude reduction in energy consumption. The energy efficiency comparison table in our manuscript specifically compares our work with other hybrid CMOS-memristor SNN chips that have undergone silicon verification, providing an appropriate and fair basis for comparison within this specific class of neuromorphic hardware. We also note that comparing with the energy consumption of MAC operations is challenging since the equivalent

operation in which neurons and synapses are realized using analog circuitry is not immediately comparable to the energy consumption of our system.

Comment 2.6

Is the energy consumption of the resistor devices included in the energy evaluation? If so, how does the resistor device modeled and evaluated? The authors should present the setups with more details.

Reply: Thank you for this question. The energy consumption of the memristive devices was not included in the energy evaluation, as only the CMOS chip was measured experimentally without the integrated devices. This distinction has been clarified in Section 2.5, where we have added more details about the experimental setup for power measurements, including:

1. Specification that measurements were performed only on the CMOS chip.
2. Breakdown of what the digital and analog circuitry account for.
3. Clarification that learning circuitry was turned off during these experiments.

This additional information should provide a clearer understanding of the scope and methodology of our power measurements. We would also like to add that such detailed power consumption measurements are largely absent from the literature regarding mixed-signal SNN processors, making this work a valuable reference point for future comparisons.

Comment 2.7

What is the benefit of supporting both STDP and SDSP learning rules?

Reply: We thank the reviewer for highlighting this important point that needed clarification. This question has helped us better articulate the motivations behind our design choice. As stated, our implementation supports both STDP and SRDP learning rules, which serve an important purpose in the context of neuromorphic computing. We have added a clarifying statement (Section 2.2) to the manuscript that explains how this combination allows for different learning mechanisms to be explored in our on-chip SNN implementation. Specifically, we note that STDP provides advantages for temporal pattern recognition and enables efficient learning with sparse spiking activity (as now demonstrated in Section 2.6), while SRDP is particularly well-suited for rate-based learning and evidence accumulation tasks. This approach enhances the versatility of the TEXEL chip as a research platform, allowing it to address a broader range of computational problems and learning paradigms that might be encountered in real-world applications.

Comment 2.8

The proposed platform is claimed to be an SNN processor, but its functionality is not demonstrated in SNN applications. At least one SNN application implemented on the proposed chip should be presented with accuracy or other metrics.

Reply: We appreciate the reviewer's valuable suggestion regarding the demonstration of SNN functionality. We fully agree that showing a concrete SNN application would significantly strengthen our work. In response, we have implemented an SNN on-chip, with detailed results presented in Section 2.6 and Section 4.7. This implementation bridges two important research areas: ultra-low-power neuromorphic computing and the VSA computational framework, making our demonstration particularly relevant to current research trends [6-8]. Our demonstration uses the processor's on-chip learning capabilities to learn distributed high-dimensional patterns representing sets of semantic items. After the learning phase, we probe the network to demonstrate its ability to correctly classify input samples into their respective sets. Importantly, this classification is achieved with minimal energy overhead (utilizing spiking input and output) and low latency, two key advantages often highlighted for mixed-signal neuromorphic chips like the one presented in this work.

Comment 2.9

Figure 3b and 3c are confusing for readers. What is the difference in the setups for the two figures? More explanations in the caption or additional legends would be appreciated.

Reply: We thank the reviewer for this valuable observation. The distinction between the experimental setups was indeed unclear in our original submission. To address this issue, we have expanded the figure caption to include a more comprehensive description of the differences between the two experiments, which should help mitigate any confusion for readers.

Replies to reviewer 3

Comment 3.1

Although a large number of synaptic plasticity functions such as STDP, SRDP and LIF can be fully reproduced in the circuit system, the above plasticity is only a single synaptic or neuronal function in the biological system, and can be achieved on a single analog memristor /RRAM cell. What are the advantages of achieving plasticity through this circuit system compared to a single memory device? Can more complex SRDP functions such as triplet STDP and BCM learning rules, be implemented by this more complex circuit chip?

Reply: We agree with the reviewer that there are extremely interesting results that showcase spike-based plasticity and neural function using single device approaches. However, while single device systems and memristive-based synapse and neuron emulators can achieve remarkable performance figures and produce impressive results, typically they have more restrictions compared to their CMOS counterparts. For example, it is not always possible to tune the spike rates produced by artificial neurons and adjust them to desired ranges (e.g., in the tens of Hertz for bio-signal processing applications or KHz for auditory processing). Similarly, the single-device synapse and neuron emulators typically have lower dynamic range for integrating inputs and producing output currents (for synapses) and firing rates (for neurons) that change linearly with the input over a large range, as can be done with CMOS analog circuits that operate in the weak-inversion domain. In addition, the CMOS synapse circuit presented in the manuscript has the ability to scale the readout currents of the memristive devices both up (e.g. to amplify the readout currents of FTJs from femtoamperes [9] to pico/nanoamperes), and down (e.g. to rescale milliamperes from ReRAM devices to nanoamperes). Therefore, a significant advantage of the CMOS implementation is its versatility and ability to be applied to a wide range of different applications, depending on the required signal regimes. The synaptic circuitry allows for the modification of synaptic time constants and therefore real time dynamics of the system. Furthermore, the implemented plasticity circuitry enables more complex learning mechanisms. For example, the second-order Calcium (Ca^{2+}) trace can be utilised as a stop learning mechanism for always on learning capabilities [10], as demonstrated in Figure 3.

Comment 3.2

The authors claim that two - or three-terminal devices can be integrated based on the system for on-chip learning. The authors need to clarify which memory devices have more energy-efficient learning based on this system. Based on this circuit system, the authors also need to give a detailed list of memristive materials to obtain the best performance.

Reply: Thank you for this important question about device compatibility. The TEXEL chip was designed with flexibility to support both two-terminal and three-terminal memristive devices. From an energy efficiency perspective, the optimal devices would be those requiring minimal energy for state switching while allowing reliable reading at low voltages. We have expanded our discussion of device compatibility in the manuscript with additional simulations. Additionally, the Supplementary Material provides a more comprehensive analysis of compatibility with specific memristive materials. To address your specific concern about energy efficiency, we have added details on how device selection impacts overall system efficiency in Section 2.5, particularly highlighting that devices requiring less energy to switch states and that can be read with small read voltages would provide the most energy-efficient learning operation.

Comment 3.3

It is well known that the energy consumption of a synapse to complete a synaptic event is only 1-10fJ. However, the energy consumption provided by the author in Table 1 is 25.9PJ, which is much higher than the energy consumption of biological systems and other chip systems. Therefore, it is necessary for the authors to improve the existing energy consumption of the circuit.

Reply: We thank the reviewer for this valuable comment, which gives us the opportunity to provide further clarification. We fully agree that biological systems are capable of performing highly complex tasks, such as vision, motor control, and adaptation to novel or noisy environments, while consuming as little as 20 Watts of power. Inspired by this remarkable efficiency, the TEXEL chip, similar to previous mixed-signal neuromorphic processors [3, 11], is designed to emulate key principles of biological neural networks, incorporating several brain-inspired features that significantly reduce the system’s overall power consumption.

The TEXEL chip integrates both analog and event-driven asynchronous digital electronic circuits on the same substrate to emulate the continuous-time local and all-or-none discrete long-range dynamics of biological neurons. Moreover, neuron and synapse circuits in the TEXEL chip function as both storage and processing units, co-localizing in the same space memory and computation. This contrasts with traditional architectures, which suffer from the memory–processing separation, leading to the well-known von Neumann bottleneck. Lastly, analog circuits in the TEXEL chip are designed with transistor operated in the sub-threshold regime, where their current flow is governed by diffusion, similar to the behaviour of protein channels in biological neurons. These features, embedded in the TEXEL chip’s architecture, have proven crucial in reducing power consumption to below the milliwatt range in various applications on previously developed neuromorphic processors (e.g., ECG [12] and HFO [13] signal processing).

Additionally, circuit-level optimizations, particularly at the neuron and synapse level, are essential to achieving ultra-low-power operation. The TEXEL neuron design draws inspiration from the improvements detailed in [2] and represents the latest step in an ongoing design evolution focused on optimizing the neuron energy efficiency. Our design implements an AdExpLIF neuron model, a widely adopted and well-understood spiking neuron model.

Specifically, the TEXEL neuron circuit consumes only tens of picojoules per spike at biological firing rates (<100 Hz). This is an order of magnitude more efficient than a similar design using the same technology node [3], and comparable to a design implemented in more advanced nodes [4]. This energy efficiency reflects several key circuit-level optimizations, detailed in [2]. Furthermore, when compared to a neuron design employing switched-capacitance techniques [5], the TEXEL neuron achieves a two-order-of-magnitude reduction in energy consumption.

To further improve energy efficiency, one could adopt smaller technology nodes [2], which would help reduce static power consumption. Alternatively, operating the neurons at lower supply voltages is another viable strategy. However, this latter approach introduces challenges, such as the need for level shifters to ensure spike compatibility with asynchronous communication circuits.

References

- [1] M. V. Nair, L. K. Muller, and G. Indiveri, “A differential memristive synapse circuit for on-line learning in neuromorphic computing systems”, *Nano Futures*, vol. 1, no. 3, p. 035 003, 2017. DOI: [10.1088/2399-1984/aa954a](https://doi.org/10.1088/2399-1984/aa954a).
- [2] A. Rubino, C. Livanelioglu, N. Qiao, M. Payvand, and G. Indiveri, “Ultra-low-power FDSOI neural circuits for extreme-edge neuromorphic intelligence”, *IEEE Transactions on Circuits and Systems I: Regular Papers*, vol. 68, no. 1, pp. 45–56, 2021. DOI: [10.1109/TCSI.2020.3035575](https://doi.org/10.1109/TCSI.2020.3035575).
- [3] S. Moradi, N. Qiao, F. Stefanini, and G. Indiveri, “A scalable multicore architecture with heterogeneous memory structures for dynamic neuromorphic asynchronous processors (DYNAPs)”, *IEEE Transactions on Biomedical Circuits and Systems*, vol. 12, no. 1, pp. 106–122, 2018. DOI: [10.1109/TBCAS.2017.2759700](https://doi.org/10.1109/TBCAS.2017.2759700).
- [4] N. Qiao and G. Indiveri, “Scaling mixed-signal neuromorphic processors to 28 nm fd-soi technologies”, in *2016 IEEE Biomedical Circuits and Systems Conference (BioCAS)*, 2016, pp. 552–555. DOI: [10.1109/biocas.2016.7833854](https://doi.org/10.1109/biocas.2016.7833854).
- [5] C. Mayr, J. Partzsch, M. Noack, S. Hanzsche, S. Scholze, S. Hoppner, G. Ellguth, and R. Schuffny, “A biological-realtime neuromorphic system in 28 nm cmos using low-leakage switched capacitor circuits”, *IEEE Transactions on Biomedical Circuits and Systems*, vol. 10, no. 1, pp. 243–254, 2016. DOI: [10.1109/tbcas.2014.2379294](https://doi.org/10.1109/tbcas.2014.2379294).

- [6] A. Renner, L. Supic, A. Danielescu, G. Indiveri, B. A. Olshausen, Y. Sandamirskaya, F. T. Sommer, and E. P. Frady, “Neuromorphic visual scene understanding with resonator networks”, *Nature Machine Intelligence*, vol. 6, no. 6, pp. 641–652, 2024. DOI: [10.1038/s42256-024-00848-0](https://doi.org/10.1038/s42256-024-00848-0).
- [7] D. Kleyko, M. Davies, E. P. Frady, P. Kanerva, S. J. Kent, B. A. Olshausen, E. Osipov, J. M. Rabaey, D. A. Rachkovskij, A. Rahimi, and F. T. Sommer, “Vector Symbolic Architectures as a Computing Framework for Emerging Hardware”, *Proceedings of the IEEE*, vol. 110, no. 10, pp. 1538–1571, 2022. DOI: [10.1109/JPROC.2022.3209104](https://doi.org/10.1109/JPROC.2022.3209104).
- [8] M. Cotteret, H. Greatorex, A. Renner, J. Chen, E. Neftci, H. Wu, G. Indiveri, M. Ziegler, and E. Chicca, “Distributed representations enable robust multi-timescale symbolic computation in neuromorphic hardware”, *Neuromorphic Computing and Engineering*, vol. 5, no. 1, p. 014 008, 2025. DOI: [10.1088/2634-4386/ada851](https://doi.org/10.1088/2634-4386/ada851).
- [9] S. Majumdar, “Back-end CMOS compatible and flexible ferroelectric memories for neuromorphic computing and adaptive sensing”, *Advanced Intelligent Systems*, vol. 4, no. 4, p. 2 100 175, 2022. DOI: <https://doi.org/10.1002/aisy.202100175>.
- [10] M. Graupner and N. Brunel, “Calcium-based plasticity model explains sensitivity of synaptic changes to spike pattern, rate, and dendritic location”, *Proceedings of the National Academy of Sciences*, vol. 109, no. 10, pp. 3991–3996, 2012. DOI: [10.1073/pnas.1109359109](https://doi.org/10.1073/pnas.1109359109).
- [11] O. Richter, C. Wu, A. M. Whatley, G. Köstinger, C. Nielsen, N. Qiao, and G. Indiveri, “DYNAP-SE2: A scalable multi-core dynamic neuromorphic asynchronous spiking neural network processor”, *Neuromorphic Computing and Engineering*, vol. 4, 2024. DOI: [10.1088/2634-4386/ad1cd7](https://doi.org/10.1088/2634-4386/ad1cd7).
- [12] F. C. Bauer, D. R. Muir, and G. Indiveri, “Real-time ultra-low power ECG anomaly detection using an event-driven neuromorphic processor”, *IEEE Transactions on Biomedical Circuits and Systems*, vol. 13, no. 6, pp. 1575–1582, 2019. DOI: [10.1109/tbcas.2019.2953001](https://doi.org/10.1109/tbcas.2019.2953001).
- [13] M. Sharifshazileh, K. Burelo, J. Sarnthein, and G. Indiveri, “An electronic neuromorphic system for real-time detection of high frequency oscillations (HFO) in intracranial EEG”, *Nature Communications*, vol. 12, no. 1, 2021. DOI: [10.1038/s41467-021-23342-2](https://doi.org/10.1038/s41467-021-23342-2).

The authors have answered most questions. However, the advancement of this work is still doubtful. The main limitation is that the demonstration of both CMOS circuits and BEOL NVM devices is not meeting the expectation of sufficient completeness. There have been several published M3D works [1][2] that have complete demonstration of CMOS circuits and other BEOL devices such RRAM, CNT, etc. Although these works do not focus on SNN applications, they are still more advanced than the CMOS-only demonstration. Considering the reported energy efficiency of this work is not outstanding, I suggest the authors submit to other venues other than Nature Communications.

[1] Y. Li et al., “Monolithic three-dimensional integration of RRAM-based hybrid memory architecture for one-shot learning,” *Nat. Commun.*, vol. 14, no. 1, p. 7140, 2023.

[2] Y. Zhang et al., “Monolithic 3D Integration of Multi-Layer CNT-CMOS/RRAM Macros for Mixed-Precision Analog-Digital Computing-in-Memory Architecture,” in *2024 IEEE International Electron Devices Meeting (IEDM), 2024*, pp. 1–4.

Reply:

We thank the reviewer for their questions and comments, which have contributed significantly to the improvement of our work. In response to their continued questions about the contributions of this research, we hope to highlight its twofold impact: First, our work aims to steer and guide research into novel devices by providing a tangible demonstration of the properties required for compatibility with full-scale systems. Second, we provide researchers fabricating devices with an accessible chip on which they can validate their work. This helps close the gap between SNN neuromorphic chips and the utilization of a diverse family of emerging devices and materials—each with different and useful properties that can eventually make an impact on such systems. Regarding the referenced articles, we agree that 3D integration is a very promising direction for improving the scalability and synaptic density of such systems. To acknowledge this, we have included a discussion with references to this direction, highlighting these works as important future avenues, particularly with regard to neuromorphic systems.

Replies to reviewer 3

The authors have addressed my major concerns. In my opinion, it may be published.

Reply: We would like to thank the reviewer for their time and their comments that helped significantly improve the manuscript.